# Characteristics of Cancer Patients in the World Trade Center Environmental Health Center

**DOI:** 10.3390/ijerph17197190

**Published:** 2020-10-01

**Authors:** Nedim Durmus, Yongzhao Shao, Alan A. Arslan, Yian Zhang, Sultan Pehlivan, Maria-Elena Fernandez-Beros, Lisette Umana, Rachel Corona, Sheila Smyth-Giambanco, Sharon A. Abbott, Joan Reibman

**Affiliations:** 1Department of Medicine, NYU Grossman School of Medicine, New York, NY 10016, USA; Nedim.Durmus@nyulangone.org (N.D.); sultan.pehlivan@nyulangone.org (S.P.); mariaelena.fernandez-beros@nyulangone.org (M.-E.F.-B.); 2Department of Population Health, NYU Grossman School of Medicine, New York, NY 10016, USA; yongzhao.shao@nyulangone.org (Y.S.); alan.arslan@nyulangone.org (A.A.A.); yian.zhang@nyulangone.org (Y.Z.); 3Department of Environmental Medicine, NYU Grossman School of Medicine, New York, NY 10016, USA; 4World Trade Center Environmental Health Center, NYC Health + Hospitals, New York, NY 10016, USA; lisette.umana@nyulangone.org (L.U.); rachel.corona@downstate.edu (R.C.); sheila.smyth-giambanco@nychhc.org (S.S.-G.); sharon.abbott@nychhc.org (S.A.A.); 5Department of Obstetrics and Gynecology, NYU Grossman School of Medicine, New York, NY 10016, USA; 6Perlmutter Comprehensive Cancer Center, NYU Langone Health, New York, NY 10016, USA

**Keywords:** World Trade Center, WTC survivors, environmental exposure, cancer, cancer characteristics

## Abstract

The destruction of the World Trade Center (WTC) towers on 11 September 2001 released many tons of aerosolized dust and smoke with potential for carcinogenic exposures to community members as well as responders. The WTC Environmental Health Center (WTC EHC) is a surveillance and treatment program for a diverse population of community members (“Survivors”), including local residents and workers, present in the NYC disaster area on 9/11 or in the days or weeks following. We report a case series of cancers identified in the WTC EHC as of 31 December 2019. Descriptive characteristics are presented for 2561 cancer patients (excluding non-melanoma skin cancer) and 5377 non-cancer WTC-EHC participants who signed informed consent. We identified a total of 2999 cancer diagnoses in 2561 patients: 2534 solid tumors (84.5%) and 465 lymphoid and hematopoietic tissue cancers (15.5%) with forty-one different cancer types. We describe the distribution, frequency, median age of cancer diagnosis and median latency from 9/11 by cancer site. In addition to common cancer types, rare cancers, including male breast cancers and mesotheliomas have been identified. The current study is the first report on cancer characteristics of enrollees at WTC EHC, a federally designated treatment and surveillance program for local community members affected by the 9/11 terrorist attack on the WTC.

## 1. Introduction

The destruction of the World Trade Center (WTC) towers on 11 September 2001 released many tons of aerosolized dust and smoke affecting the surrounding community [1,2]. Community members, including those who lived, worked, attended schools or cleaned many of the contaminated sites, had potential for adverse health effects from acute exposures to the dust clouds created by the collapsing buildings, as well as chronic exposures to the resuspended dust or fumes from the fires that burned through December 2001 [3,4,5,6]. Components of the dust and fumes included respirable particulate matter (PM10 and PM2.5), containing a combination of pulverized cement, glass fibers, asbestos, lead, and combustion products, as well as complex mixtures of volatile chemicals including polycyclic aromatic hydrocarbons, polychlorinated biphenyls, and polychlorinated furans and dioxins [7,8,9,10,11,12,13,14,15]. Programs to monitor and treat the adverse health effects in the responders and community members were developed in the years after 9/11. These programs were eventually combined as the WTC Health Program (WTCHP) under the auspices of the Centers for Disease Control and Prevention (CDC), National Institutes of Safety and Occupational Health (NIOSH) [3] and became law (under the James Zadroga 9/11 Health and Compensation Act H.R. 847) in 2010 for implementation in 2011. The WTCHP serves four distinct groups, the New York City Fire Department (FDNY) responders, the WTC general responders, the Pentagon/Shanksville responders, and community members (“Survivors”), each with their own Clinical Centers of Excellence (CCE). The WTC Environmental Health Center (WTC EHC) is the CCE for the survivor cohort [3].

During the past decade, recruitment into these programs has been promoted by many methods, first and foremost through the involvement of numerous community-based organizations as well as multi-media campaigns including advertisements on the subway, radio, and newspapers. Individuals can also be referred to the WTCHP by their medical providers, lawyers, or community advocates. Enrollment is voluntary and must be self-initiated. Community members, called “Survivors”, must document their location and activities on 11 September 2001, as well as time periods and hours spent in the 1.5 m radius of the former WTC complex in the days and weeks following the disaster. As of 31 December 2019, 11,038 individuals were enrolled in the WTC EHC, a fraction of the estimated 500,000 people living or working in the WTC disaster area on 11 September 2001 [6,16].

Cancer development is a complex process, and tumor progression occurs by a sequence of randomly occurring changes in genetic material by a change in cell functions such as proliferation, survival, and growth inhibition, which may take a long time [17,18,19]. The minimum time period (i.e., latency) for cancer patients who have been exposed to WTC dust must have elapsed between the initial date of the individual’s 9/11 exposure and the date of the initial diagnosis of the individual’s cancer for the cancer to be certified. Minimum latency periods for specific types or categories of cancer were described by policy [20].

Estimates of minimum latencies are available in the scientific literature for only a small number of the covered cancers associated with exposure to carcinogenic agents present in the aftermath of the 9/11 attacks (also referred to as “9/11 agents”) [20]. Early studies suggested an unusual occurrence of multiple myeloma in a younger age group in the WTC responders [21] and an excess incidence (19%) for all cancer sites combined among the WTC-exposed firefighters following 9/11 [3], with prostate and thyroid cancers contributing to the excess rate. In view of these data, and the presence of multiple carcinogenic agents in the WTC dust and fumes, suggesting biologic plausibility, the WTC Health Program included most cancers as “Certifiable conditions” in October 2012 with the latency rules for specific cancer’s inclusion defined by the WTCHP [20].

WTC Health Registry was established by the New York City Department of Health as an epidemiologic study to monitor and document long-term physical and mental health effects of 9/11. Participants were enrolled in the WTC Health Registry between September 2003 and November 2004 and enrollment to the WTC Health Registry is now closed. The WTC Health Registry does not provide health care. WTC EHC is a treatment and surveillance program, enrollment to WTC EHC requires certifiable physical and/or mental health disorders related to 9/11. Individuals can enroll in both the WTC Health Registry (HR) and WTC EHC. There have been reports about WTC-related cancers from FDNY and other WTC responder programs as well as cancers in affected community members by the WTC Health Registry [2,7,21,22,23,24,25,26]. However, there has been no formal reports describing cancers at the WTC EHC. To fill this knowledge gap, we describe the characteristics of all reported cancers after 9/11 in the WTC EHC to facilitate future studies on cancers in community members exposed to an environmental disaster. To enhance our ability to characterize cancers, we relied on the extensive demographic, exposure, and clinical information obtained during surveillance in a clinical WTC EHC database and created an additional cancer-dedicated database with detailed information on all cancer types (WTC EHC Pan-Cancer database) [27]. Our goal was to improve our ability to characterize individual cancer types, common characteristics across cancer types, and support subsequent studies of cancers among local community members. We now report a case series of the cancers identified in the WTC EHC and included in the WTC EHC Pan-Cancer database as of 31 December 2019.

## 2. Materials and Methods

### 2.1. Patient Enrollment

The WTC Environmental Health Center (WTC EHC) was created in response to community requests in the years after 9/11 and was included as the Center of Excellence in the WTCHP for local community members, subsequently called “Survivors” [3]. Patients self-refer to this program and under law, in contrast to rules for the responder programs, enrollment requires the presence of a “Certifiable condition” [3,4]. Individuals can be enrolled on the WTC Health Program for an initial health evaluation based on exposure criteria defined by the CDC-NIOSH [28] that includes geographic boundaries and; (1) presence in the dust or dust cloud in the New York City disaster area on 11 September 2001; (2) work, residence, or school/daycare attendance in the New York City disaster area; (3) work as a cleanup worker or maintenance work in the New York City disaster area; (4) documented residence during the period beginning on 11 September 2001, and ending on 31 May 2003; (5) employment in the defined geographic area at any time during the period beginning on 11 September 2001, and ending on 31 May 2003 [29]. In addition, the enrollee must have a certifiable WTC-related health condition that includes an aerodigestive disorder, mental health symptoms, consistent with PTSD, depression or anxiety, or cancer [29]. Thus, enrollees with and without cancers are included, and children under the age of 18 on 9 November 2001 are also eligible [30].

Monitoring of patients in the WTC EHC remains ongoing, whereas we have ongoing treatment visits to also monitor patients for routine evaluations every 12–18 months. At these visits, patients undergo standardized medical and mental health evaluations. Any new cancer diagnoses are documented and verified.

Our initial and monitoring questionnaires collect information on basic exposures including, occupational and lifestyle exposures (e.g., smoking) as well as WTC exposure. We do not obtain extensive data on family histories; however, we obtain consent allowing us to re-contact patients to obtain more extensive information when indicated.

### 2.2. Exposure Assessment at WTC EHC

Community members had very complex exposures, which have been the source of extensive discussion for many years. Community members may have been in the area on 9/11, and subsequently had massive dust exposure on 11 September 2001 (acute exposure, which we describe as being exposed to the dust clouds on 9/11 (dust cloud yes/no)). In addition, community members may have had chronic exposure to resuspended dust as residents—few of whom were evacuated, or as local workers, most of whom returned approximately one week after the disaster to incompletely cleaned areas (defined as chronic exposure). WTC EHC participants also included workers involved in the cleaning of the surrounding area (clean-up workers) and a small number of responders who did not fit the responder program for a variety of reasons. For our first analysis, we simplified categories of exposure as those with “acute” exposure e.g., they were there on 9/11 and in the dust cloud (dust cloud: yes). We then include potential for chronic exposure, which depends on the category of activity i.e., local resident, local worker, student, clean-up worker. These categories are not mutually exclusive, participants may have both dust cloud exposure as well as potential for chronic exposure as a local resident, local worker etc. Previous studies have shown that the categories of potential chronic exposure as resident, local worker, clean-up worker, and others are significantly associated with pulmonary disorders, neuropathic disorders, and mental health disorders in the WTC EHC participants [1,3,4,5,6,8,9,10,12,15].

### 2.3. Clinical and Cancer Characteristics in the WTC EHC

All patient characteristics are obtained from the WTC EHC clinical database. This database [3] includes basic demographic information, WTC and other exposures, the clinical characteristics and details of exposures gathered via administered questionnaires completed at the time of enrollment and monitoring.

The cancer characteristics of patients are derived from the newly developed WTC EHC Pan-Cancer database, which was created to capture information on all cancer types in the WTC EHC and to interface with current clinical databases [27]. Briefly, our new pan-cancer database uses research electronic data capture (REDCap) as a secure Federal Information Security Management Act (FISMA) and The Health Insurance Portability and Accountability Act (HIPAA)-compliant environment, geared to support online or offline data capture for research studies [31]. Study data are collected and managed using REDCap electronic data capture tools hosted at New York University. REDCap is a secure, web-based software platform designed to support data capture for research studies, providing; (1) an intuitive interface for validated data capture; (2) audit trails for tracking data manipulation and export procedures; (3) automated export procedures for seamless data downloads to common statistical packages; and (4) procedures for data integration and interoperability with external sources [31,32].

The WTC EHC Pan Cancer database includes cancers that are “Certifiable” within the WTCHP rules [20]. As per the WTCHP criteria, cervical and uterine cancers are non-certifiable unless they are invasive. The list of the cancers that have been included in our pan-cancer database has been shown in Appendix A. In this paper, we included only primary cancer diagnoses confirmed by a pathology/cytology report and used the date of the first diagnostic pathology/cytology report as the date of diagnosis. The age of diagnosis and latency period in years after 9/11 have been calculated by using these data. Information on cancer characteristics is obtained from clinical and pathology reports, medical record reviews, and state tumor registries and includes documentation of source. Cancer type (ICD-10 code), age at diagnosis, laterality, location of initial pathology report, multiplicity (number of tumors at primary site), tumor size, post-surgical tumor margin status, grade, tumor histology (ICD-O-3 code), TNM classification of the tumor, and stage are all included. Standard tumor-specific information (e.g., Gleason score for prostate cancer, specific staging for lymphoid and hematopoietic tissue cancers) is recorded. The number of patients who had more than one primary cancer is recorded. Multiple primaries were defined according to the International Association of Cancer Registries and International Agency for Research on Cancer (IACR/IARC) criteria [33].

For this analysis, we include patients with a diagnosis of cancer who were enrolled in the WTC EHC between May 2002 and 31 December 2019. For reference of sampling, we also included WTC EHC participants who had signed research consent and had a medical condition that did not include a diagnosis of cancer. These non-cancer participants were enrolled at one clinical site (Bellevue Hospital) during the same period and were used to compare cancer and non-cancer participants to understand if there were basic characteristics or exposures that differed between these groups. Although most patients with cancers were enrolled in the WTC EHC with a cancer as their certifying illness, some patients developed incident cancers after enrollment for a non-cancer condition.

The study was approved by the New York University School of Medicine Institutional Review Board (IRB). Reference group of non-cancer participants who have signed consent for analysis of their data (IRB number: i06-1) were included in the study. Patients with cancer were analyzed after removal of personal identifiers with IRB approval to review de-identified data (IRB number: i06-1_MOD49). Documentation of consent to be re-contacted is included for subsequent studies.

### 2.4. Analysis

The study population and cancer and non-cancer subgroups are displayed using a flow chart. Descriptive statistics were used to summarize population characteristics including median and range to summarize data from continuous variables and counts and percentages to summarize data from binary or categorical variables. Demographic characteristics and WTC exposure status were numerically summarized for the cancer group and the non-cancer group, respectively. Frequency distributions of top cancers were graphically displayed using flow charts and bar graphs for female and male, respectively. Statistical software R-3.6.3 was utilized to conduct these analyses.

## 3. Results

### 3.1. Participants

We identified 11,038 patients who were enrolled in the WTC EHC between May 2002 and 31 December 2019. Among them, 2840 patients had a diagnosis of any type of cancer. For this analysis, we excluded patients with non-melanoma skin cancers (n = 279), leaving 2561 patients with 2999 cancer diagnoses (Figure 1).

Characteristics of participants with or without cancer are shown in Table 1. Nearly half of the cancer patients (46.7%) and non-cancer participants (49.8%) were women. The median age on 11 September 2001 was comparable in cancer and non-cancer participants (46.2 and 41.6 years, respectively). The distribution of race and ethnicity was diverse and included minority groups: African-Americans, Asians, Hispanics, and Native Americans (Table 1). Nearly 50% of the patients reported acute exposure having been caught in the WTC dust cloud on 9/11 and the majority of them (51.6% among non-cancer participants and 64.0% among cancer patients) were local workers. Most participants with or without cancer were never smokers (65% and 69.2%, respectively).

### 3.2. Cancer Types

Of the 2561 patients diagnosed with a cancer with a diagnostic pathology/cytology report, 13% (n = 335) had more than one primary cancer diagnosis at different time periods. As shown in Table 2, 11.7% (n = 299) of cancer patients had two primary cancer diagnoses and 36 patients had three or more primary cancers, including four patients with four and one patient with five different primary cancer diagnoses. Over 13% of breast cancer patients in our population had a second primary cancer (13.6%) and 17% of prostate cancer patients had multiple primary cancers. Among lung cancer patients, 24.3% had multiple primary cancers. Over 17% of our patients with thyroid cancer had multiple primary cancers.

Of the 2999 cancer diagnoses identified in the 2561 patients, forty-one different cancer sites were identified. Types of the diagnosed cancers are listed in Appendix A. Three cancers had an unknown primary site. Most cancers were solid tumors (n = 2534, 84.5%) and the rest were lymphoid and hematopoietic tissue cancers (n = 465, 15.5%). Among lymphoid and hematopoietic tissue cancers, most (n = 217, 7.3%) were lymphomas, followed by leukemias, myelomas, myeloproliferative neoplasms (MPN) and myelodysplastic syndromes (MDS) (Figure 2).

The top fifteen most common cancer diagnoses are shown in Figure 3. Overall, breast (22%) and prostate (16%) cancer diagnoses were the most common, followed by lung (9%), thyroid (7%) and lymphoma (7%) (Figure 3).

Several rare cancers were also identified, including nine male breast cancer patients (two of them with two primary breast cancer diagnoses), two patients with peritoneal mesothelioma, and one patient with lung mesothelioma.

### 3.3. Cancer Diagnoses and Sex

Breast cancer (46%) was the most common cancer diagnosis in women, followed by lung (11%), thyroid (9%), and lymphoma (6%) (Figure 4A). In men, prostate cancer was the most common diagnosis (30%), followed by lymphoma (8%), lung (7%) and head and neck site cancers (7%) (Figure 4B).

As shown in Figure 5, a total of 465 lymphoid and hematopoietic tissue cancer diagnoses were identified. Cancers of lymphoid and hematopoietic tissue were more frequently diagnosed in men (64%) than in women (36%). Lymphoma was the most common lymphoid and hematopoietic tissue cancer type in both men (28%) and women (19%). Leukemia and myeloma diagnoses were more common in men (17% and 13%, respectively) compared with those found in women (7% and 9%, respectively).

### 3.4. Cancer Diagnoses and Age

The median age in years at first cancer diagnosis and median latency in years from 9/11 for the top fifteen cancer types and mesotheliomas are shown for the total population as well as for female and male patients (Table 3). The median age at diagnosis across all cancer sites was 59 years (range from 5 to 95 years). The median latency for all cancer types was 12.8 years (range from 0.7 to 18.3 years). The median age and latency in years for less common cancer types are shown in Appendix A.

## 4. Discussion

In this study, we described the distribution of the cancer types in a cohort of community members (“Survivors”) who self-referred to the WTC EHC in New York City. The study population was racially and ethnically diverse, and almost half were women. Breast cancer is the most common cancer diagnosed in the WTC EHC, followed by prostate, thyroid and combined hematopoietic and lymphoid cancers. Differences in cancer distribution were detected between men and women and rare cancers, including male breast cancers and mesotheliomas, have been identified. We also describe high rates of patients with multiple primary cancers.

The dust, debris, and fumes from the WTC disaster contained known and suspected carcinogens, including polycyclic aromatic hydrocarbons, asbestos, benzene, and dioxins [3,13,34,35,36], all of which are associated with multiple cancer types. Previous studies reported increased cancer rates in rescue and recovery workers exposed to the WTC dust and fumes [7,22,23,25]. A pooled cohort of 29,993 US firefighters showed excess cancer mortality and incidence [37], with significant increases in incidence of digestive and respiratory cancers, and mesothelioma. In an extended follow-up study, excess incidence of prostate and thyroid cancers was reported among rescue and recovery workers, while small but statistically significantly higher than expected rates were found for skin melanoma in both rescue and recovery workers and civilians, and female breast cancer and non-Hodgkin’s lymphoma among civilians [22]. However, there are few studies of cancers in the local community members (“Survivors”). This paper reports characteristics of the cancers in a NIOSH designated clinical program for local community members.

We developed a pan-cancer database to facilitate understanding of the relationship between environmental exposures and cancers. Information on cancer characteristics is obtained from a careful review of pathology reports, medical records or state tumor registries with documentation of source included [27]. Cancers are identified from self-report from currently enrolled patients, newly self-referred patients enrolling in the program, as well as from linkages with state cancer registries. Importantly, according to the NIOSH guidelines and standards, patients enrolled in the WTC EHC are required to have defined “certifiable” physical and/or mental health conditions [27]. The availability of both cancer and cancer-free subjects in this study gave us an opportunity to investigate the possible differences between these groups, both of whom have been exposed to WTC dust. We describe a large variety of cancer types identified in the WTC EHC program. In contrast to the overall distribution of cancer types in studies of rescue and recovery workers, we report on racially and ethnically diverse population, almost half of which were women. Breast cancer is the most common cancer diagnosed in the WTC EHC, followed by prostate, thyroid and combined hematopoietic and lymphoid cancers. This finding is consistent with data from the WTC Health Registry analysis of cancers in community members [22]. Our distribution of cancers is likely influenced by the large number of women in the WTC EHC, which is in contrast to the predominantly male responder programs. Among men, prostate and lymphoid and hematopoietic tissue cancers were the most common cancers in the WTC EHC, similar to that described for the predominantly male rescue and recovery workers [7,22,23,24,25,26,37].

We also describe high rates of multiple primary cancers in the WTC EHC. Over 13% of patients had multiple primary malignancies. Over 13% of breast cancer patients in our population had a second primary cancer (13.6%) and 17% of prostate cancer patients had multiple primary cancers. Among lung cancer patients, 24.3% had multiple primary cancers. Multiple primary cancers were reported for thyroid cancer in responders [26], which is consistent with our data.

We, as well as others, describe a large variety of cancers (41 types of cancers) as well as multiple cancers per patient among WTC-exposed local community members. The large variety of cancer types is consistent with that described in other WTC-exposed populations. The data from our pan-cancer database will allow us to investigate the involvement of key carcinogenic mechanisms and processes in the future by investigating the specific cancer characteristics and related biomarker information.

Inclusion of women in the WTC EHC allowed us to analyze cancer characteristics in this understudied group. We describe differences in cancer characteristics among women and men. Importantly, prostate cancer, lymphomas and lung cancers were the most common cancer in men, whereas breast, lung and thyroid cancers were the most common cancers in women.

Age on 11 September 2001 showed a statistically significant association with cancer risk, with a 1.09-fold greater risk for each one-year increase in general and 1.13-fold increase in prostate cancer in responders [24]. When compared to the Surveillance, Epidemiology, and End Results (SEER) program data [38] and cancer statistics published by the American Cancer Society [39], there is a suggestion that many cancers in the WTC EHC are presenting at an earlier than expected age, whereas some cancer types, such as testicular cancers, are presented at older than expected age. The diverse distribution of race/ethnicity and sex in our population suggests that future studies of individual cancers in the WTC EHC program should consider these variables along with other potential confounders.

This study has important strengths. This is one of the few descriptions of cancers in WTC-exposed local community members rather than those involved in rescue and recovery activities. The diverse population of nearly 50% of women allows for the description of different cancer frequencies in women compared to men and will allow for future studies of WTC-related health effects in women. The description of a large variety of solid as well as lymphoid and hematopoietic tissue malignancies supports the diverse nature of the malignancies and the need to study these cancers as a group (pan-cancer analysis) as well as individually. As such, this study sets the stage for future studies of environmental exposure and cancer latency, cancer characteristics, and underlying mechanisms of cancer development. The finding of rare as well as common cancers in this population may provide clues about environmental exposures and the underlying mechanisms of the development of these cancers. The potential for continued follow-up of these patients will provide further insight into cancer behavior and prognosis.

There are several limitations to our study. Our population is a self-referred population and, therefore, subject to selection bias and also only cancers named as “certifiable” by the WTCHP could be included. For this reason, we cannot directly assess cancer incidence or mortality rates in this population. Our patients have been enrolled in WTC health program and they are being offered screening within this program routinely. This screening may explain the detection of some but not all cancers since many enroll with previously diagnosed cancer and we do not report overall prevalence of cancers. We may have some missing cancer diagnoses since some of our patients had cancers diagnosed before the WTCHP allowed certification and some of these patients may have died from these early cancers before we identified them for this study.

## 5. Conclusions

We provide an initial description of cancers in local community members with exposure to the WTC dust and fumes. This description provides an overview of the cancers diagnosed in a civilian cohort and will facilitate future analyses of this population as well as clinical management. The pathologic and histologic characteristics of cancers with multiple primary cancer analysis for each type of cancer, assessment of cancer risk factors and cancer-specific biomarker information in our WTC EHC Pan-Cancer database will allow for future analyses of individual and group cancer behaviors in this group, as well as long-term follow-up study of this population. These studies have the potential to provide information on cancer characteristics seen in comparison to the general population, and their relationship to the WTC exposures.

## Figures and Tables

**Figure 1 ijerph-17-07190-f001:**
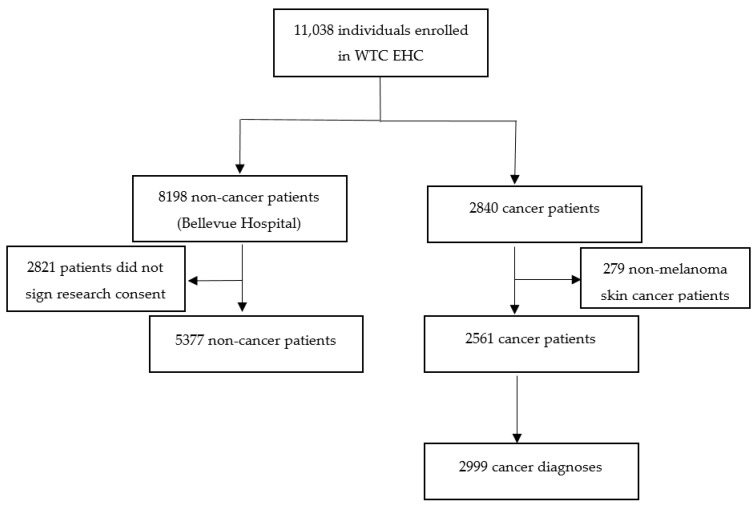
World Trade Center Environmental Health Center (WTC EHC) patients included in the WTC EHC cancer analysis as of 31 December 2019.

**Figure 2 ijerph-17-07190-f002:**
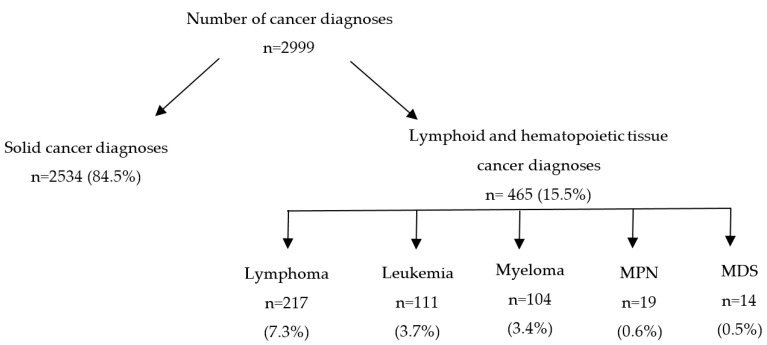
Distribution of solid, lymphoid and hematopoietic tissue cancer diagnoses in the WTC EHC (MPN = myeloproliferative neoplasms; MDS = myelodysplastic syndromes).

**Figure 3 ijerph-17-07190-f003:**
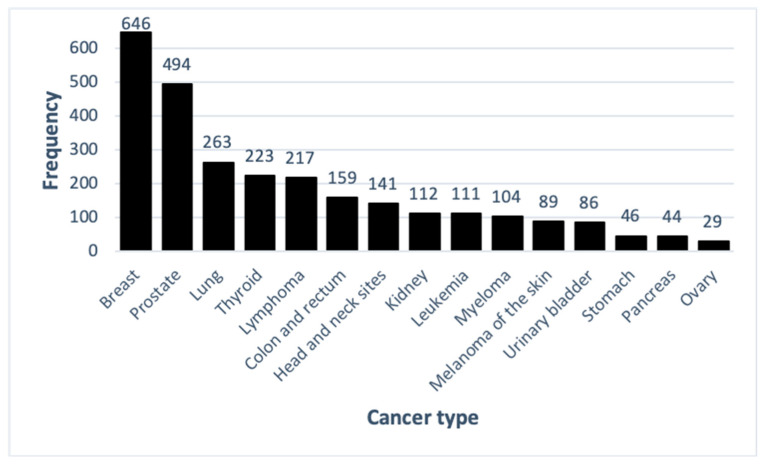
Frequency of top fifteen cancer diagnoses in both sexes.

**Figure 4 ijerph-17-07190-f004:**
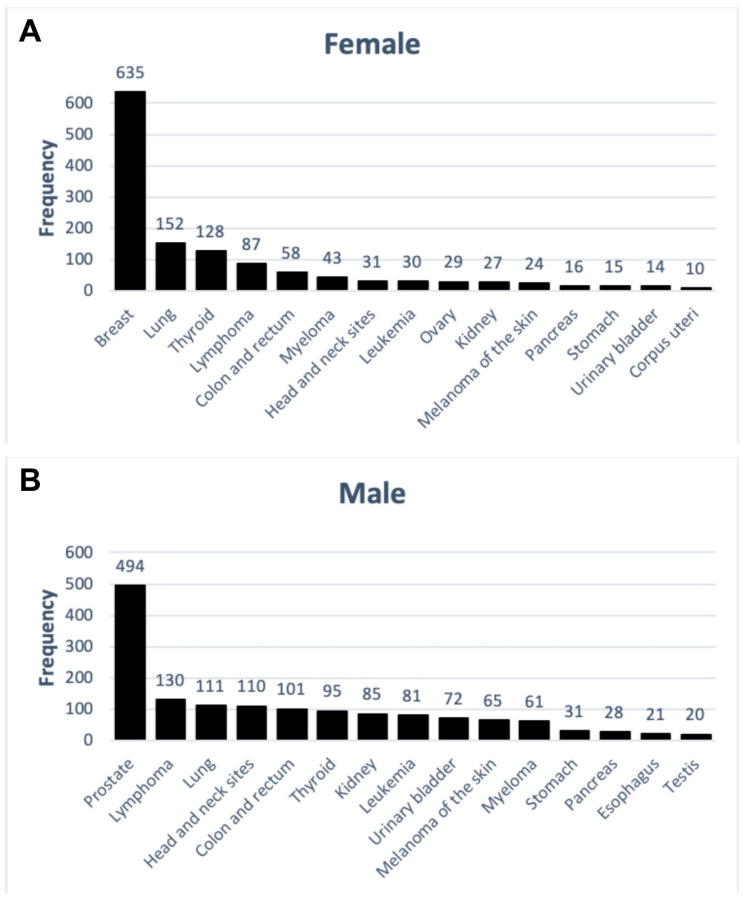
Frequency of top fifteen solid cancer diagnoses in female (**A**) and male (**B**) patients.

**Figure 5 ijerph-17-07190-f005:**
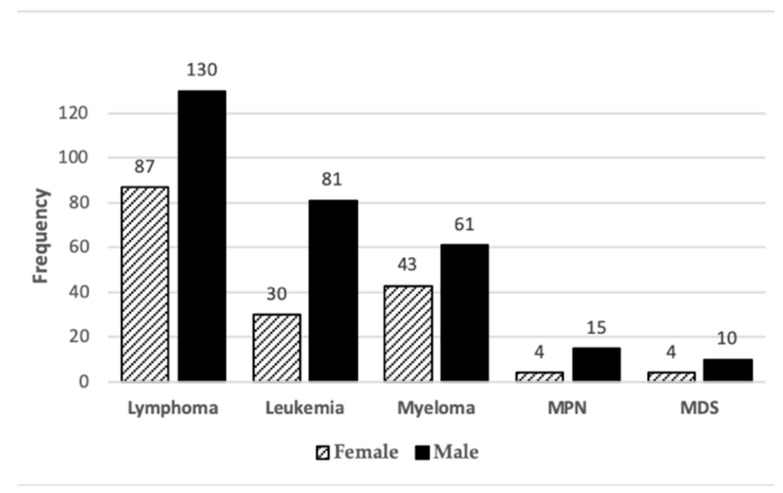
Frequency of cancers of lymphoid and hematopoietic tissue in female and male patients in WTC EHC (MPN = myeloproliferative neoplasms; MDS = myelodysplastic syndromes).

**Table 1 ijerph-17-07190-t001:** Characteristics of patients with/without cancers at enrollment in the WTC EHC.

		Non-Cancer	Cancer
**n**		5377	2561
**Sex, n (%)**	F	2678 (49.8)	1195 (46.7)
	M	2699 (50.2)	1366 (53.3)
**Age on 9/11 (years), median [range]**		41.6 [0.0, 82.6]	46.2 [0.0, 81.9]
**Age on 9/11 (years), n (%)**	≤20	309 (5.7)	28 (1.1)
	21–29	613 (11.4)	169 (6.6)
	30–39	1454 (27.0)	510 (19.9)
	40–49	1781 (33.1)	913 (35.7)
	50–59	963 (17.9)	755 (29.5)
	60–69	202 (3.8)	172 (6.7)
	70–79	50 (0.9)	13 (0.5)
	≥80	5 (0.1)	1 (0.0)
**Race/Ethnicity, n (%)**	Hispanic	1592 (30.3)	246 (11.8)
	NH *-White	1992 (37.9)	1099 (52.9)
	NH *-Black	1051 (20.0)	394 (19.0)
	Asian	610 (11.6)	334 (16.1)
	Native American	14 (0.3)	3 (0.1)
**BMI **, n (%)**	Normal weight (<25)	1365 (28.7)	486 (30.3)
	Overweight (25–30)	1717 (36.1)	616 (38.4)
	Obese (≥30)	1671 (35.2)	504 (31.4)
**Income, n (%)**	≤$30,000/year	2842 (54.8)	860 (42.4)
	>$30,000/year	2340 (45.2)	1169 (57.6)
**Education, n (%)**	High school or less	1801 (33.5)	578 (27.3)
	More than high school	3572 (66.5)	1541 (72.7)
**Exposure information**			
**Dust cloud exposure, n (%)**	No	2506 (47.0)	1018 (48.1)
	Yes	2824 (53.0)	1098 (51.9)
**Type of community members** **, n (%)**	Local Worker Resident Student	2766 (51.6) 1329 (24.8) 132 (2.5)	1352 (64.0) 607 (28.8) 40 (1.9)
	Clean-up worker	619 (11.5)	17 (0.8)
	Other	516 (9.6)	95 (4.5)
**Smoking, n (%)**	Never (≤1 p-y ***)	3661 (69.2)	1363 (65.0)
	Former (>1 p-y)	1076 (20.3)	621 (29.6)
	Current (>1 p-y)	555 (10.5)	114 (5.4)
**Pack year, n (%)**	≤5 p-y	4167 (78.7)	1513 (72.1)
	>5 p-y	1125 (21.3)	585 (27.9)

* NH: non-Hispanic, ** the unit for BMI is kg/m^2^, *** *p*-y: pack year.

**Table 2 ijerph-17-07190-t002:** Distribution of primary cancer diagnoses per cancer patient.

Number of Cancers	n	Percent (%)
*1*	2226	86.90
*2*	299	11.70
*3*	31	1.20
*4*	4	0.20
*5*	1	0.04
*Total*	2561	100

**Table 3 ijerph-17-07190-t003:** The median age of diagnosis and median latency in years from 9/11 distribution for top fifteen cancer and mesothelioma in male and female in the WTC EHC.

	All (n = 2999)			Male (n = 1623)			Female (n = 1376)		
Cancer Type	n (%)	Median Age of Diagnosis (Range) (Year)	Median Latency (Range) (Year)	n (%)	Median Age of Diagnosis (Range) (Year)	Median Latency (Range) (Year)	n (%)	Median Age of Diagnosis (Range) (Year)	Median Latency (Range) (Year)
Breast	646 (22)	55 (27, 86)	12.4 (3.3, 17.8)	11 (1)	58 (41, 82)	11.5 (4.7, 16.6)	635 (46)	55 (27, 86)	12.4 (3.3, 17.8)
Prostate	494 (16)	62 (40, 83)	12.3 (3.9, 18.2)	494 (30)	62 (40, 83)	12.3 (3.9, 18.2)			
Lung	263 (9)	64 (34, 89)	14 (3.3, 18.1)	111 (7)	66 (38, 89)	14.4 (5.3, 18.1)	152 (11)	63 (34, 85)	13.7 (3.3, 18.1)
Thyroid	223 (7)	52 (13, 80)	11.8 (2.7, 17.9)	95 (6)	52 (22, 75)	12.4 (2.7, 17.7)	128 (9)	49 (13, 80)	11.4 (3.0, 17.9)
Lymphoma	217 (7)	57 (21, 91)	12 (0.9, 17.8)	130 (8)	56 (21, 82)	11.8 (1.5, 17.5)	87 (6)	59 (22, 91)	12.7 (0.9, 17.8)
Colon and rectum	159 (5)	59 (29, 84)	13.4 (4.1, 18.1)	101 (6)	59 (36, 84)	13.7 (4.1, 18.1)	58 (4)	59 (29, 82)	12.4 (4.9, 17.2)
Head and neck sites	141 (5)	58 (26, 87)	12.9 (3.6, 18.1)	110 (7)	58 (37, 87)	11.9 (3.6, 18.1)	31 (2)	57 (26, 73)	13.2 (7.7, 17.1)
Kidney	112 (4)	58 (7, 80)	12.8 (4.2, 17.7)	85 (5)	58 (7, 80)	12.7 (4.2, 17.7)	27 (2)	58 (32, 73)	13.2 (6.4, 17.6)
Leukemia	111 (4)	59 (16, 81)	12.4 (0.7, 16.8)	81 (5)	59 (16, 81)	12.2 (0.7, 16.7)	30 (2)	58 (34, 72)	12.6 (2.1, 16.8)
Myeloma	104 (3)	60 (28, 82)	13.9 (3.9, 17.8)	61(4)	59 (28, 79)	13.2 (5.2, 17.8)	43 (3)	60 (40, 82)	14.3 (3.9, 17.7)
Melanoma of the skin	89 (3)	59 (34, 85)	12.7 (3.2, 17.6)	65 (4)	60 (37, 85)	12.8 (3.2, 17.6)	24 (2)	55 (34, 74)	9.7 (4.2, 16.9)
Urinary bladder	86 (3)	65 (34, 95)	13.2 (4.2, 17.4)	72 (4)	65 (34, 95)	13.3 (4.2, 17.4)	14 (1)	64 (35, 75)	12.2 (6.6, 16.8)
Gonad (Testis/Ovary)	49 (2)	52 (28, 82)	12.2 (4.3, 17.6)	20 (1)	41 (28, 70)	8.8 (4.3, 16.9)	29 (2)	57 (28, 82)	13.1 (8.2, 17.6)
Stomach	46 (2)	59 (39, 75)	13.8 (4.4, 17.8)	31 (2)	57 (39, 72)	13.9 (4.4, 17.8)	15 (1)	60 (49, 75)	12.4 (4.4, 17.0)
Pancreas	44 (1)	64 (46, 83)	14.8 (5.7, 17.7)	28 (2)	64 (49, 83)	14.9 (6.9, 17.7)	16 (1)	60 (46, 79)	14.2 (5.7, 17.5)
Mesothelioma	3 (0)	57 (38, 70)	15.4 (15.0, 16.7)	2 (0)	54 (38, 70)	16 (15.4, 16.7)	1 (0)	57 (57, 57)	15 (15.0, 15.0)

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
