# Peer review of "Characteristics of Cancer Patients in the World Trade Center Environmental Health Center"

_ijerph, 2020, doi:10.3390/ijerph17197190_

Round 1

Reviewer 1 Report

Durmus et al report the results of a case-series, describing the characteristics of cancer cases reported after 9/11 in the WTC Environmental Health Center in order to characterise cancer types and characteristics. 

Minor revisions

On page 3, line 123 and in figure 1, the authors report to have 2,561 patients in the final study population, however in the abstract and in table 1 they report results on 2,840 cancer patients (including the non-melanoma skin cancer patients). Could you please clarify why table 1 includes all cancer patients?

Table 4: Could you please add (years) to the tables to indicate that latency is reported in years?

Author Response

We would like to thank the reviewers for their detailed and careful review for our manuscript entitled “Characteristics of cancer patients in the world trade center environmental health center” by Durmus et al. (ID ijerph-900789). We now provide a point-by-point response to the comments of the reviewers. We believe that these revisions strengthen the manuscript, and we are very appreciative of their comments. We believe that our revised manuscript is of value to this issue of the journal, which focuses on adverse health effects of WTC exposures.

Reviewer 1

Durmus et al report the results of a case-series, describing the characteristics of cancer cases reported after 9/11 in the WTC Environmental Health Center in order to characterize cancer types and characteristics. 

Minor revisions

Comment: On page 3, line 123 and in figure 1, the authors report to have 2,561 patients in the final study population, however in the abstract and in table 1 they report results on 2,840 cancer patients (including the non-melanoma skin cancer patients). Could you please clarify why table 1 includes all cancer patients?

Answer: We identified 11,038 patients who were enrolled in the WTC EHC between May 2002 and December 31, 2019. Among them, 2,840 patients had a diagnosis of any type of cancer. In the analysis for this paper, we excluded patients with non-melanoma skin cancers (n = 279), leaving 2,561 patients with 2,999 cancer diagnoses. We show these numbers in Figure 1. We have revised Table 1 as recommended and excluded patients with non-melanoma skin cancer. Revised Table 1 has been added to the paper.

Comment: Table 4: Could you please add (years) to the tables to indicate that latency is reported in years?

Answer: The explanation indicating that the latency is in years has been added to both table legend and also table. Table 3 and Table 4 has been merged per suggestion of another reviewer.

Reviewer 2 Report

Very well written.  Two minor comments.

First, line 183 gives 46.5 years as the median age of diagnosis across all cancer sites.  However, this seems impossible, since the lowest age of diagnosis in Table 3 for any cancer site is 52.

Second, line 141 refers to patents with a cancer diagnosis but does not state what constitutes a diagnosis.  Readers with a nonspecialist background may wonder what that includes.  For example, suppose a person got a diagnosis as a second opinion.  Would that be included?

Author Response

We would like to thank the reviewers for their detailed and careful review of our manuscript entitled “Characteristics of cancer patients in the world trade center environmental health center” by Durmus et al. (ID ijerph-900789). We now provide a point-by-point response to the comments of the reviewers. We believe that these revisions strengthen the manuscript, and we are very appreciative of their comments. We believe that our revised manuscript is of value to this issue of the journal, which focuses on adverse health effects of WTC exposures.

Reviewer 2

Very well written.  Two minor comments.

Comment: First, line 183 gives 46.5 years as the median age of diagnosis across all cancer sites.  However, this seems impossible, since the lowest age of diagnosis in Table 3 for any cancer site is 52.

Answer: We thank to the reviewer for attention and apologize for the incorrect number shown. The median age at diagnosis across all cancer sites was 59 years (range from 5 to 95 years).  We have corrected this information on Page 11 Line 312.

Comment: Second, line 141 refers to patents with a cancer diagnosis but does not state what constitutes a diagnosis.  Readers with a non-specialist background may wonder what that includes.  For example, suppose a person got a diagnosis as a second opinion.  Would that be included?

Answer: We thank the reviewer for pointing out an important point. We used the diagnosis and diagnosis date from first diagnostic pathology/cytology report. We have now clarified this and provided more detailed information about this information on Page 5 Line 194.

Reviewer 3 Report

Peer review report:

Summary:

This case-series study aimed to describe socio-demographic and clinical characteristics of cancer patients identified in the World Trade Center Environmental Health Center as of 31 December 2019. The WTC EHC is a surveillance and treatment program for community members present in the NYC disaster area on 9/11. The manuscript provides the first descriptive findings of cancer patients enrolled in the WTC EHC, which comprise diverse populations in terms of sex and ethnicity, and presents data on WTC dust exposure. This preliminary report will facilitate the conduct of future studies on cancers in the survivors of the WTC terrorist attack. Publication in the special issue of the International Journal of Environmental Research and Public Health on “9/11 health update” is relevant.

Broad comments:

Strengths:

- The study presents new and extended data on community members (survivors) affected by the disaster; clinical and sociodemographic data are described, as well as exposure data.

- Context and aims are well addressed. Results, discussion and conclusions are reasonable considering that this is a first report.

Limitations:

- There is potential selection bias as the population is self-referred and cancer patients are identified as such if they were diagnosed with a certifiable cancer; this has been mentioned. Can you provide the completion of the program compared to the estimated number of community members affected by the disaster?

- Can more results be provided, for instance, a more detailed description of the cancer patients with more than one cancer diagnosis, cancer risk factors and biomarker information? Could you provide a complete list of the 41 types of cancers (in supplementary material for instance)?

- Cancer patients have been compared to a reference group of non-cancer participants, but what was the objective of such comparison? This hasn’t been discussed.

- The methods should be more detailed, in particular the exposure assessment.

- Some information is misplaced throughout the manuscript.

Comments and questions:

Methods:

- Enrolment in the WTC EHC is not detailed. Was there some kind of campaign to enrol in the WTC EHC? How did people know about it? Were participants referred by their physician, and why (cancer diagnosis?)? What are the eligibility criteria for inclusion in the WTC EHC?

- What is the completion of the program compare to the estimated expected number of community members affected by the attack?

- Regarding the study population for this descriptive analysis, what are the eligibility criteria for inclusion? Students were part of the study population, were they over 18 years old? Does the WTC EHC include children?

- Ascertainment of cases is missing. How was cancer occurrence reported, it is self-reported? Were all cancer diagnoses confirmed by histopathological reports? Were there incident cases, meaning that participants were free of cancer at enrolment in the WTC EHC?

- Regarding monitoring in the WTC EHC, how is it performed? Do participants regularly complete questionnaires? If yes, how frequently? Do you have data on other cancer risk factors, such as family/personal history of cancer? Have you presented demographic data at enrolment in the WTC or at diagnosis? This should be specified in the methods.

- Exposure assessment is missing; there are no references and no description of the exposure assessment of dust cloud exposure. How was it performed? Was it self-reported? Have the participants been geocoded? In the methods, both acute and chronic exposures are detailed (lines 88-89), but not in the results. Why acute and chronic exposures have not been described? Are they both included in the “dust exposure”? If yes, how?

- What was the attempt with the comparison of cancer patients with a non-cancer group? Why no statistical tests have been provided for the comparison? The reference group (“non-cancer patients”) is not defined. Why did they enrol in the WTC EHC? What was the reason they self-reported to the WTC EHC? Do they have another illness than cancer? I would not use the term “patients” for them.

- In the paragraph 2.2 Analysis, what did you mean by “cancer biomarker information”? It seems that it hasn’t been described in the results.

Results:

- There shouldn’t be information about the methods in the results (for instance lines 124-126).

- Globally in the results, there seems to be no missing data. Is there no missing data in your database or have you selected participants with no missing data? This should be mentioned in the methods.

- The term “exposure category” in Table 1 seems not appropriate as this is not an exposure; I would suggest “type of community members”.

- Could you describe the characteristics of patients with more than one cancer diagnosis (for instance, age at each diagnosis and location)? Are they all primary cancer? This should be mentioned in the methods.

- Could you describe the delay between enrolment in the WTC EHC and cancer diagnosis?

Discussion:

- First paragraph of the discussion should present the main findings of the study rather than the context and objectives.

- What is the difference between local community members and civilians (lines 213-214)? What is the difference between the WTC health registry and the WTC EHC? Can individuals enrolled in the registry also be enrolled in the EHC?

- Is there screening programs for breast and prostate cancer in NYC? Do you know if screening could explain a part of the elevated numbers of breast and prostate cancers?

- Results are presented for the first time in the discussion; it should be moved to the results (patients with more than one cancer diagnosis).

- For the comparison with the American cancer society data and testicular cancer: what type of testicular cancer are you talking about? Do men with GCNIS-related TGCT get cancer later than 45 years? Or is it another type of testicular cancer?

References:

- Ref 18: Has this been published? The reference is not complete.

- Ref 32 is not complete.

No acknowledgments?

Specific comments

Abstract: some redundancy between lines 25 to 27 and lines 34 to 36.

Introduction:

Line 53: (Survivors)?

Lines 65 to 67: the sentence is somewhat unclear.

Line 67: FDNY should to be explained.

Lines 67-69: please provide some references.

Some redundancy between lines 70-72 and 75-77.

Methods:

Line 84: Citing the use of REDCap: https://projectredcap.org/resources/citations/

> Please cite the publications below in study manuscripts using REDCap for data collection and management. We recommend the following boilerplate language:

Study data were collected and managed using REDCap electronic data capture tools hosted at [YOUR INSTITUTION].1,2 REDCap (Research Electronic Data Capture) is a secure, web-based software platform designed to support data capture for research studies, providing 1) an intuitive interface for validated data capture; 2) audit trails for tracking data manipulation and export procedures; 3) automated export procedures for seamless data downloads to common statistical packages; and 4) procedures for data integration and interoperability with external sources.

1PA Harris, R Taylor, R Thielke, J Payne, N Gonzalez, JG. Conde, Research electronic data capture (REDCap) – A metadata-driven methodology and workflow process for providing translational research informatics supportJ Biomed Inform. 2009 Apr;42(2):377-81.

2PA Harris, R Taylor, BL Minor, V Elliott, M Fernandez, L O’Neal, L McLeod, G Delacqua, F Delacqua, J Kirby, SN Duda, REDCap Consortium, The REDCap consortium: Building an international community of software partnersJ Biomed Inform. 2019 May 9 [doi: 10.1016/j.jbi.2019.103208]

Line 84: FISMA and HIPAA should be explained.

Lines 87-88: What are the other exposures (than WTC)? Can this be detailed?

Results:

Line 121, line 124: the term “patients” seems not suitable here > “individuals” or “participants” seem more appropriate.

Lines 124-126: this information is found in the results but it should be move to the methods.

Table 1:

- Why present age characteristic as continuous and categorical, I would advise to choose one. Add the unit for age.

- NH should be explained.

- Unit for BMI is missing. Normal BMI is 18-25 kg/m²

- Pack year unit: p-y or p-yr?

- Number of decimals should be standardized.

Table 2

- Can you add other characteristics to describe these cancer patients?

- Number of decimals should be standardized.

Figure 3, figure 4 and table 3 present the same data (column ALL and by sex). I suggest keeping data from table 3.

There is some redundancy between tables 3 and 4, which both present n (%) of cancer type. Can table 3 and 4 be regrouped into one table?

Table 3:

- Number of decimals should be standardized.

Figure 5:

- I would suggest to present percentages in column (by sex) rather than in line. Also, figure 5 presents frequency; therefore, it is not easy to retrieve the percentages presented in the text.

Discussion:

- Line 201 missing a dot.

- Lines 198-199: some redundancy between first and 2nd paragraph of discussion (lines 198-199 and 207-208).

- Lines 219-221: should be moved to the methods.

- Lines 221-225: these results were not presented in the results part. This should be moved to results.

- Line 243: suggestion “the American cancer society”.

Author Response

We would like to thank the reviewers for their detailed and careful review of our manuscript entitled “Characteristics of cancer patients in the world trade center environmental health center” by Durmus et al. (ID ijerph-900789). We now provide a point-by-point response to the comments of the reviewers. We believe that these revisions strengthen the manuscript, and we are very appreciative of their comments. We believe that our revised manuscript is of value to this issue of the journal, which focuses on adverse health effects of WTC exposures.

Summary:

This case-series study aimed to describe socio-demographic and clinical characteristics of cancer patients identified in the World Trade Center Environmental Health Center as of 31 December 2019. The WTC EHC is a surveillance and treatment program for community members present in the NYC disaster area on 9/11. The manuscript provides the first descriptive findings of cancer patients enrolled in the WTC EHC, which comprise diverse populations in terms of sex and ethnicity, and presents data on WTC dust exposure. This preliminary report will facilitate the conduct of future studies on cancers in the survivors of the WTC terrorist attack. Publication in the special issue of the International Journal of Environmental Research and Public Health on “9/11 health update” is relevant. 

Broad comments:

Strengths:

- The study presents new and extended data on community members (survivors) affected by the disaster; clinical and sociodemographic data are described, as well as exposure data.

- Context and aims are well addressed. Results, discussion and conclusions are reasonable considering that this is a first report. 

Limitations:

Comment: There is potential selection bias as the population is self-referred and cancer patients are identified as such if they were diagnosed with a certifiable cancer; this has been mentioned. Can you provide the completion of the program compared to the estimated number of community members affected by the disaster?

Answer: We thank to the reviewer for this comment. Our population is a self-referred population and, therefore as pointed out by the reviewer, subject to selection bias. In addition, only cancers deemed “certifiable” by the WTCHP are eligible for inclusion. We have described this inclusion criteria and potential for bias on page 4 line 135 . In addition, we have stated on page 14 line 412 that because of the recruitment criteria, we cannot assess cancer incidence or mortality rates in this population. We may also have some missing cancer diagnoses since some of our patients had cancers diagnosed before the WTCHP allowed cancer certification and some of these patients may have died from these early cancers before we identified them for this study. We have now clarified this in the description of study limitations in the Discussion section page 14 line 416. Despite this bias, we believe that a description of the cancers in this group is important contribution to the literature.

Exact number of exposed community members with potential for adverse health effects from dust/fume exposure is unknown. Farfel et al. (2009), estimated potentially 500,000 people living or working in the WTC Disaster area on 9/11/2001. As of December 31, 2019, 11,038 individuals were enrolled in the WTC EHC.

This is a long-term program and the enrollment is still ongoing. In this paper, we describe the characteristics of cancer cases enrolled as of December 31, 2019. All enrolled members are eligible for an initial health screening visit at one of three designated clinical sites.  Following the initial visit, ongoing monitoring and treatment visits for Survivors are dependent on the presence of a certified” health condition as defined by the WTCHP and this is an ongoing screening program.

Monitoring of patients in the WTC EHC remains ongoing. Whereas we have ongoing treatment visits, the goal is to also monitor patients for routine evaluations every 12 - 18 months. At these visits, patients undergo standardized medical and mental health evaluations. Any new cancer diagnoses are documented and verified. Detailed information about this added to Page 4  line 141Comment: Can more results be provided, for instance, a more detailed description of the cancer patients with more than one cancer diagnosis, cancer risk factors and biomarker information? Could you provide a complete list of the 41 types of cancers (in supplementary material for instance)?

Answer:  The reviewer asks a very important question about detailed cancer characteristics. The manuscript provides the first descriptive findings of cancer patients enrolled in the WTC EHC, which will facilitate the conduct of future studies on cancers in the survivors of the WTC terrorist attack including analyses to understand the characteristics of many specific cancers in this population and will lead to more detailed analysis including studies of those with multiple cancers, cancer risk factors and biomarkers. This manuscript describes the rich dataset that will be available for these future studies.

The analysis about multiple primary cancers, cancer risk factors and biomarker information are still ongoing.

The list of the included 41 types of cancers has been added as suggested in the new supplemental table (Table S1). Explanation about this has been added to Methods section on Page 5, Line 192.

Comment: Cancer patients have been compared to a reference group of non-cancer participants, but what was the objective of such comparison? This hasn’t been discussed.

Answer: As of December 31, 2019, 11,038 individuals were enrolled in the WTC EHC and 2,840 patients had any type of cancer. Since this is our first description of the individuals in our program with cancer, contrasting the groups in the WTC EHC population with and without cancer is a basic type of epidemiological analysis of interest to understand if there were basic characteristics or exposures that differed between these two groups. This point has been discussed on Page 5, Line 207.

Comment: The methods should be more detailed, in particular the exposure assessment.

Answer: The reviewer asks an important question about exposure assessment in the community. In contrast to responders, community members had very complex exposures, which have been the source of extensive discussion for many years. Community members may have been in the area on 9/11 themselves, and subsequently had massive dust exposure on that day (acute exposure, which we describe as being exposed to dust cloud on 9/11). In addition, community members may have had chronic exposure to resuspended dust as residents – few of whom were evacuated, or as local workers, most of whom returned approximately one week after the disaster to incompletely cleaned areas (defined as chronic exposure). WTC EHC participants also included leaning workers and some responders. The responders later moved to other programs after the formation of WTCHP. Thus, the categories of chronic exposure are classified as exposure categories including residence, local workers, students, clean-up workers, and others. Importantly, the acute (dust cloud exposure on 9/11/2001), and categories of chronic and acute/chronic exposure (residence, local worker, clean-up worker, others) have been shown to be significantly associated with pulmonary disorders, neuropathic disorders, and mental health disorders in the WTC EHC participants [1,3-6,8-10,12,15]. Therefore, we routinely classify WTC exposure categories as acute, chronic or acute/chronic. In this study, we also used dust cloud exposure information. Additional detailed exposure histories such as occupational and lifestyle exposures (e.g. smoking) were also obtained by questionnaires at first enrollment to the program.

We added a paragraph explaining this in the Methods section on page 4 line 158.Comment: Some information is misplaced throughout the manuscript.

Answer: Manuscript has been reviewed by our team regarding this issue and misplaced information have been rearranged.Comments and questions:

Methods:

Comment: Enrolment in the WTC EHC is not detailed. Was there some kind of campaign to enroll in the WTC EHC? How did people know about it? Were participants referred by their physician, and why (cancer diagnosis?)? What are the eligibility criteria for inclusion in the WTC EHC?

Answer: The program to monitor and treat the adverse health effects in the community members (“Survivors”) was developed in the years after 9/11. This program began as a self-funded, subsequently philanthropic-funded, then city and finally federally-funded program, which was eventually combined with the Responder program as the WTC Health Program (WTCHP) under the auspices of the Centers for Disease Control and Prevention (CDC), National Institutes of Safety and Occupational Health (NIOSH) and became law under the James Zadroga 9/11 Health and Compensation Act H.R. 847 in 2010 for implementation in 2011. The community members were renamed “Survivors” under the law.

The individuals in the WTCHP “Survivor” Centers of Excellence can be enrolled to WTC Health Program for an initial health evaluation based on one of the following exposure criteria: 1) The screening applicant was present in the dust or dust cloud in the New York City disaster area on September 11, 2001. 2) The screening applicant worked, resided, or attended school, childcare, or adult daycare in the New York City disaster area, for at least 4 days during the period beginning on September 11, 2001, and ending on January 10, 2002; or 30 days during the period beginning on September 11, 2001, and ending on July 31, 2002. 3) The screening applicant worked as a cleanup worker or performed maintenance work in the New York City disaster area during the period beginning on September 11, 2001, and ending on January 10, 2002, and had extensive exposure to WTC dust as a result of such work. 4) Residents who possessed a lease for a residence or purchased a residence in the New York City disaster area and resided in such residence during the period beginning on September 11, 2001, and ending on May 31, 2003. 5) Individuals whose place of employment at any time during the period beginning on September 11, 2001, and ending on May 31, 2003, was in the New York City disaster area were also deemed eligible for enrollment to the program (https://www.cdc.gov/wtc/regulations2.html).

All enrolled members are eligible for an initial health screening visit at one of three designated clinical sites.  Following the initial visit, ongoing monitoring and treatment visits for Survivors are dependent on the presence of a “certified” health condition as defined by the WTCHP. 

Comment: What is the completion of the program compare to the estimated expected number of community members affected by the attack?

Answer: As of December 31, 2019, 11,038 individuals were enrolled in the WTC EHC, a fraction of the estimated 500,000 people living or working in the WTC Disaster area on September 11, 2001. Enrollment remains open and ongoing.

Following the initial visit, ongoing monitoring and treatment visits for Survivors are dependent on the presence of a certified” health condition as defined by the WTCHP and this screening is continuing.

Monitoring of patients in the WTC EHC remains ongoing. Whereas we have ongoing treatment visits, the goal is to also monitor patients for routine evaluations every 12 - 18 months. At these visits, patients undergo standardized medical and mental health evaluations. Any new cancer diagnoses are documented and verified. Detailed information added to on Page 4 line 141.

Comment: Regarding the study population for this descriptive analysis, what are the eligibility criteria for inclusion? Students were part of the study population, were they over 18 years old? Does the WTC EHC include children?

Answer: The James Zadroga 9/11 Health and Compensation Act of 2010 (H.R. 847) was passed by the US Congress, a law designed to respond to the adverse health effects of the disaster, was authorized in December 2010 and reauthorized in 2015. This Act coordinates the clinical and surveillance programs that had been initiated by community groups and later included in the WTC Health Program (WTCHP) under CDC-NIOSH (Centers for Disease Control and Prevention - National Institute for Occupational Safety and Health). Any person who has defined WTC exposure and a community member who has the presence of a certifiable WTC-related health condition including students and children under 18  have been enrolled in the treatment program. The enrollment criteria are cited at the following URL https://www.cdc.gov/wtc/regulations2.html/

Any person who has defined WTC exposure and a community member who has the presence of a certifiable WTC-related health condition including students and children under 18 have been enrolled in the treatment program. The WTC EHC has a pediatric section and includes children. All enrolled members are eligible for an initial health screening visit at one of three designated clinical sites. Following the initial visit, ongoing monitoring and treatment visits for Survivors are dependent on the presence of a certified” health condition as defined by the WTCHP.

Explanation about enrollment has been added to Page 4, Line 135.

Comment: Ascertainment of cases is missing. How was cancer occurrence reported, it is self-reported? Were all cancer diagnoses confirmed by histopathological reports? Were there incident cases, meaning that participants were free of cancer at enrolment in the WTC EHC?

Answer: Soon after the terrorist attack, there were concerns about potentially elevated cancer risk among WTC Survivors and Responders exposed to a complex mix of toxic chemicals. The James Zadroga 9/11 Health and Compensation Act of 2010 (H.R. 847) was passed by the US Congress, a law designed to respond to the adverse health effects of the disaster, was authorized in December 2010 and reauthorized in 2015. As increased cancer rates began to be described in WTC Responders and WTC Survivors, many cancers were added to the list of WTC-related diseases in 2012.

Individuals may be referred to the WTCHP by their self-referral, medical providers, lawyers, or community advocates, but enrollment is voluntary and must be self-initiated.   The cancer cases include incidence cases who were free of cancer at enrollment in the WTC EHC. Information on cancer characteristics was obtained from clinical and pathology reports, medical record reviews and state tumor registries. The source of information is documented in the database.  

In this paper, we included only cancer diagnoses confirmed by a pathology/cytology report and used the date of the first diagnostic pathology/cytology report as the date of diagnosis

This information has been added to the Page 5 Line 194.

Comment: Regarding monitoring in the WTC EHC, how is it performed? Do participants regularly complete questionnaires? If yes, how frequently? Do you have data on other cancer risk factors, such as family/personal history of cancer? Have you presented demographic data at enrolment in the WTC or at diagnosis? This should be specified in the methods.

Answer: Monitoring of patients in the WTC EHC remains ongoing. Whereas we have ongoing treatment visits, the goal is to also monitor patients for routine evaluations every 12 - 18 months. At these visits, patients undergo standardized medical and mental health evaluations. Any new cancer diagnoses are documented and verified.

This information added to Page 4, Line 141.

Our initial and monitoring questionnaires collect information on basic exposures including occupational and lifestyle exposure history including smoking history. Unfortunately, we do not have extensive data on family histories. We do obtain consents allowing us to re-contact these patients and our goal will be obtained more extensive information in subsets of cancer patients.

This information added to Page 5, Line 171.

All demographic data presented are from the questionnaires at the time of enrollment.

This information added to Page 4, Line 150.

Comment: Exposure assessment is missing; there are no references and no description of the exposure assessment of dust cloud exposure. How was it performed? Was it self-reported? Have the participants been geocoded? In the methods, both acute and chronic exposures are detailed (lines 88-89), but not in the results. Why acute and chronic exposures have not been described? Are they both included in the “dust exposure”? If yes, how?

Answer: Community members had very complex exposures, which have been the source of extensive discussion for many years. Community members may have been in the area on 9/11 themselves, and subsequently had massive dust exposure on that day (acute exposure, which we describe as dust cloud). In addition, community members may have had chronic exposure to resuspended dust as residents (few of whom were evacuated), or as local workers, most of whom returned approximately 1 week after the disaster to incompletely cleaned areas. We therefore routinely classify WTC exposure as acute, chronic or acute/chronic.

We have now detailed this more clearly on Page 4 Line 158.

Information is obtained as self-report but documentation or attestation is required to be included in the program. We do obtain precise addresses of residences and work-places, which could be used for subsequent geocoding, however most community members moved around a lot (were evacuated, returned intermittently to homes, changed work places etc.), and performed complex activities which may have enhance or reduced exposures (cleaned homes and workplaces, etc.) making geocoding difficult but not impossible.

Comment: What was the attempt with the comparison of cancer patients with a non-cancer group? Why no statistical tests have been provided for the comparison? The reference group (“non-cancer patients”) is not defined. Why did they enroll in the WTC EHC? What was the reason they self-reported to the WTC EHC? Do they have another illness than cancer? I would not use the term “patients” for them.

Answer: We identified 11,038 patients who were enrolled in the WTC EHC between May 2002 and December 31, 2019. Among them, 2,561 patients had a diagnosis of any type of cancer (excluding non-melanoma skin cancer). For comparison, we included 5,377 WTC EHC participants who had signed research consent, but did not have a diagnosis of cancer, but had another health condition (e.g., pulmonary, digestive disorder, mental health condition) to be enrolled in the WTC EHC (Page 5. Line 207).

All subjects enrolled to WTC EHC are required by law to have certifiable physical disorders or mental disorders related to 9/11, so all of them are patients. But to prevent misunderstanding, we used the term “participant” for non-cancer patients throughout the paper.

Comment: In the paragraph 2.2 Analysis, what did you mean by “cancer biomarker information”? It seems that it hasn’t been described in the results.

Answer: The WTC EHC Pan-Cancer database includes patient demographics, cancer characteristics, and cancer biomarker information for each cancer that is available from the pathology and clinical reports. The cancer biomarkers are often cancer site-specific, thus, biomarker information for each cancer are still being analyzed. To prevent the misunderstanding the Analysis section has been modified as suggested. 

Results:

Comment: There shouldn’t be information about the methods in the results (for instance lines 124-126).

Answer: The methods information in the results section has been moved to the Methods section.

Comment: Globally in the results, there seems to be no missing data. Is there no missing data in your database or have you selected participants with no missing data? This should be mentioned in the methods.

Answer: For this descriptive study, we do not have any key descriptive information (e.g. age, sex) missing. This information has been mentioned in Methods section, Page 6, Line 215.

Comment: The term “exposure category” in Table 1 seems not appropriate as this is not an exposure; I would suggest “type of community members”.

Answer: The term “exposure category” in Table 1 has been changed to “type of community members” as suggested.

Comment: Could you describe the characteristics of patients with more than one cancer diagnosis (for instance, age at each diagnosis and location)? Are they all primary cancer? This should be mentioned in the methods.

Answer: The number of patients who had more than one primary cancer has been listed in the paper. All of the cancer in patients with more than one cancer had been reported as primary cancers. The detailed analysis of multiple primary cancers occurring in the WTC EHC participants is undergoing and will be subject of a separate paper in the future.

This information has been mentioned Page  5, Line 203.

Comment: Could you describe the delay between enrollment in the WTC EHC and cancer diagnosis?

Answer: Most patients enroll in the program with aerodigestive disorders or mental health issues. Some of these patients subsequently developed incident cancers. Many patients with cancer were unaware of the program or did not fit eligibility until they developed a cancer.  Moreover, cancer eligibility includes latency rules specific to cancer types (Page 3, Line 93).

Discussion:

Comment: First paragraph of the discussion should present the main findings of the study rather than the context and objectives.

Answer: An extra paragraph with the main findings of the paper has been added to the beginning of Discussion section.

Comment: What is the difference between local community members and civilians (lines 213-214)? What is the difference between the WTC health registry and the WTC EHC? Can individuals enrolled in the registry also be enrolled in the EHC?

Answer: “Civilian” was used to distinguish someone who was NOT a Responder (firefighter, police, medical rescue personnel, etc.), but who assisted in volunteer efforts in the disaster area.   

WTC Health Registry was established by the New York City Department of Health as an epidemiologic study to monitor and document long-term physical and mental health effects of 9/11. Participants were enrolled in the WTC Health Registry between September 2003 and November 2004 and enrollment to the WTC Health Registry is now closed.  The WTC Health Registry does not provide health care. WTC EHC is a treatment and surveillance program, enrollment to WTC EHC requires certifiable physical and/or mental health disorders related to 9/11. Individually can enroll in the both WTC HR and WTC EHC.

This information has been added to Introduction section Page 3, Line 97.

Comment: Is there screening programs for breast and prostate cancer in NYC? Do you know if screening could explain a part of the elevated numbers of breast and prostate cancers?

Answer: Our patients have been enrolled in WTC health program and they are being offered screening within this program routinely. This screening may explain the detection of some but not all cancers. This has been described in the Discussion section on Page 14, Line 412.

Comment: Results are presented for the first time in the discussion; it should be moved to the results (patients with more than one cancer diagnosis).

Answer: The results presented for the first time about multiple primary cancers have been moved to the Results section. Discussion about this subject has been modified.

Comment: For the comparison with the American cancer society data and testicular cancer: what type of testicular cancer are you talking about? Do men with GCNIS-related TGCT get cancer later than 45 years? Or is it another type of testicular cancer?

Answer: We report the testicular cancer as overall testicular cancers as per the reporting in the SEER database. The median age for all types of testicular cancers is 33 years in SEER database whereas in our study we had a median age of 41. American Cancer Society facts and figures also refer to SEER data. However, the reviewer makes a very important point that detailed patient and tumor characteristics of this cancer are important and this will be a focus of future studies. The reference link for this cancer about median age has been added to the discussion and reference list. (https://seer.cancer.gov/statfacts/html/testis.html)

 References:

Comment: Ref 18: Has this been published? The reference is not complete.

Answer: This paper has not been published yet and has been submitted and currently under peer-review.

Comment: Ref 32 is not complete.

Answer: Reference 32 has been modified and corrected.

Comment: No acknowledgments?

Answer: We thank the reviewer for this reminder. We now acknowledge all the community-based organizations and community members and patients who have contributed to the success of this program. We also would like to thank Angeles Pai and Michelle Hyde for their administrative efforts in the program and Janice Levas, Kymara Kyng, Adrienne Adessi, Renee Liberty and Susan Soa for their efforts in the cancer care of these patients.

Specific comments

Abstract:

Comment: Some redundancy between lines 25 to 27 and lines 34 to 36.

Answer: Abstract has been modified as suggested.

Introduction:

Comment: Line 53: (Survivors)?

Answer: This sentence has been modified.

Comment: Lines 65 to 67: the sentence is somewhat unclear.

Answer: This sentence has been modified.

Comment: Line 67: FDNY should to be explained.

Answer: FDNY (The New York City Fire Department) has been described in Page 2, Line 55.

Comment: Lines 67-69: please provide some references.

Answer: References have been added.

Comment: Some redundancy between lines 70-72 and 75-77.

Answer: These sentences were modified as suggested.

Methods:

Comment: Line 84: Citing the use of REDCap: https://projectredcap.org/resources/citations/

> Please cite the publications below in study manuscripts using REDCap for data collection and management. We recommend the following boilerplate language:

Study data were collected and managed using REDCap electronic data capture tools hosted at [YOUR INSTITUTION].1,2 REDCap (Research Electronic Data Capture) is a secure, web-based software platform designed to support data capture for research studies, providing 1) an intuitive interface for validated data capture; 2) audit trails for tracking data manipulation and export procedures; 3) automated export procedures for seamless data downloads to common statistical packages; and 4) procedures for data integration and interoperability with external sources.

1PA Harris, R Taylor, R Thielke, J Payne, N Gonzalez, JG. Conde, Research electronic data capture (REDCap) – A metadata-driven methodology and workflow process for providing translational research informatics supportJ Biomed Inform. 2009 Apr;42(2):377-81.

2PA Harris, R Taylor, BL Minor, V Elliott, M Fernandez, L O’Neal, L McLeod, G Delacqua, F Delacqua, J Kirby, SN Duda, REDCap Consortium, The REDCap consortium: Building an international community of software partnersJ Biomed Inform. 2019 May 9 [doi: 10.1016/j.jbi.2019.103208]

 Answer: REDCap has been cited as recommended.

Comment: Line 84: FISMA and HIPAA should be explained.

Answer: The Federal Information Security Management Act (FISMA) (Page 5, Line 180) and HIPAA (The Health Insurance Portability and Accountability Act) (Page 5, Line 181) have been explained.

Comment: Lines 87-88: What are the other exposures (than WTC)? Can this be detailed?

Answer: We include additional exposures such as occupational and lifestyle exposures (e.g. smoking) in WTC EHC database. The explanation about other exposures has been added to  Page 4, Line 145.

 Results:

Comment: Line 121, line 124: the term “patients” seems not suitable here > “individuals” or “participants” seem more appropriate.

Answer: We included WTC EHC participants who signed research consent, and were enrolled at one clinical site, (Bellevue Hospital), during the same period for non-cancer health conditions. Since these individuals are enrolled for treatment and care, they are required by law to have certifiable mental or physical disorders related to 9/11, thus we used the term patient.  We changed it to participant throughout the paper as suggested by the reviewer.

Comment: Lines 124-126: this information is found in the results but it should be move to the methods.

Answer: This information has been moved to the Methods section.

Table 1:

Comment: Why present age characteristic as continuous and categorical, I would advise to choose one. Add the unit for age.

Answer:  We appreciate the comment of the reviewer. However, the age listed in table a summary of the age of the patients on Sep 11, 2001. To evaluate the effect of age of exposure on cancer development we used age as a categorical variable with age as decades. We believe that the use of these two types of age descriptions is appropriate for the different points that we are making.  

The unit for age has been added.

Comment: NH should be explained.

Answer: NH is explained under the table as Non-Hispanic.

Comment: Unit for BMI is missing. Normal BMI is 18-25 kg/m²

Answer: Explanation for BMI unit has been added under the Table 1.

Comment: Pack year unit: p-y or p-yr?

Answer: Pack year unit has been changed as p-y to be consistent. Also, the definition has been added under the Table 1.

Comment: Number of decimals should be standardized.

Answer: Number of decimals have been standardized

 Table 2

Comment: Can you add other characteristics to describe these cancer patients?

Answer: This manuscript is an overview of basic characteristics of cancer patients at WTC EHC. We understand that there are many questions that will be asked to understand the association of other characteristics and their role as risk for cancer or cancer characteristics in future research. This manuscript facilitates such future analysis.

Comment:  Number of decimals should be standardized.

 Answer: Number of decimals have been standardized

Comment: Figure 3, figure 4 and table 3 present the same data (column ALL and by sex). I suggest keeping data from table 3.

Answer: Figure 3 shows the frequency of top fifteen cancer diagnoses in both sexes whereas Figure 4 shows frequency separated by sex. Sometimes seeing data in table and figure differs from each other and can be useful. Thus, we wanted to show the related data as table and figure (Figure 4 and Table 3).

 Comment: There is some redundancy between tables 3 and 4, which both present n (%) of cancer type. Can table 3 and 4 be regrouped into one table?

Answer: Table 3 and 4 has been merged as suggested.

 Table 3:

Comment: Number of decimals should be standardized.

Answer: Number of decimals have been standardized

Figure 5:

Comment: I would suggest to present percentages in column (by sex) rather than in line. Also, figure 5 presents frequency; therefore, it is not easy to retrieve the percentages presented in the text.

Answer: As we have presented frequency throughout the paper, to be consistent with other figures we showed the frequency in Figure 5. The percentages for each type of cancer have been summarized in the results section Page 10, Line 298.

 Discussion:

Comment:  Line 201 missing a dot.

Answer: Missing dot has been added.

Comment: Lines 198-199: some redundancy between first and 2nd paragraph of discussion (lines 198-199 and 207-208).

Answer: Redundant sentences have been removed.

Comment: Lines 219-221: should be moved to the methods.

Answer: This sentence has been moved to Methods section.

Comment: Lines 221-225: these results were not presented in the results part. This should be moved to results.

Answer: These results has been moved to Results section

Comment: Line 243: suggestion “the American cancer society”.

Answer: The sentence has been modified with a different reference.

Reviewer 4 Report

General Comments:

The manuscript presents some descriptive characteristics of about 3,000 cancer patients in the world trade centre environmental health center (WTC EHC) by comparison with 5,377 cancer-free WTC EHC patients followed up through December 31, 2019, and by cancer site. This study adds to the limited bulk of information about the health consequences, specifically cancer, in local community members affected by the 9/11 terrorist attacks on the WTC. It highlights the WTC EHC Pan-Cancer Database as an important source of information for further health surveillance and research in the population. It would be important to expand more the Materials and Methods section by explaining how information on non-WTC-related risk factors were collected (e.g., BMI, smoking status). The authors mentioned three cases of mesothelioma occurred in about 3,000 cancer patients with about 18 years of follow-up. Despite the small number it is important and strongly recommended to include description of these cases in the main results rather than in the Supplementary materials. The authors mentioned that information on clinically significant biomarkers is collected but they didn’t present any descriptive results in the manuscript, as well as there are no distinct results on cancer patients with multiple primary malignancies. The presented results do not provide sufficient ground for the subsequent detailed discussion on possible carcinogenic mechanisms as a result of WTC exposure in this population, this specific paragraph is very speculative and to reviewer’s opinion is not very relevant to the discussion of results. At this stage, the manuscript provides only some basic descriptive characteristics, it would be very interesting and important to see further advanced cancer data statistical analyses in the affected local community members.                                 

Specific comments:

1.       Results, Table 1: Why the characteristics are given for all 2,840 cancer patients although in the preceding para (p.3 lines 122-124) authors stated that 279 patients with non-melanoma skin cancer were excluded for this analysis?

2.       Results, Table 1: Please indicate when the BMI measurements were performed, especially in cancer patients as their diseases could directly affect the BMI.

3.       Results, Table 1: Same question for smoking status, at what moment of time? Cancer diagnosis could probably be the reason of a higher proportion of former smokers compared to cancer-free group.

4.       P.4, line 143: please clarify if all five cancer diagnoses in one patient were primary cancers.

5.       Figure 2. The number of myeloma cases on that figure (n=101) is different from the number of 104 myelomas in the following figures and tables, please check and correct.

6.       P.7, line 172, Title of Figure 5: it is suggested to change “blood cancer” for “cancers of lymphoid and haematopoietic tissue” according to ICD terminology and nomenclature.               

7.       Table 3 and 4: please include characteristics for mesothelioma cases. It would be interesting and important to see in the main results sex, age and latency characteristics of the three cases taking into account asbestos exposure in the study population.

Author Response

We would like to thank the reviewers for their detailed and careful review of our manuscript entitled “Characteristics of cancer patients in the world trade center environmental health center” by Durmus et al. (ID ijerph-900789). We now provide a point-by-point response to the comments of the reviewers. We believe that these revisions strengthen the manuscript, and we are very appreciative of their comments. We believe that our revised manuscript is of value to this issue of the journal, which focuses on adverse health effects of WTC exposures.

General Comments:

The manuscript presents some descriptive characteristics of about 3,000 cancer patients in the world trade center environmental health center (WTC EHC) by comparison with 5,377 cancer-free WTC EHC patients followed up through December 31, 2019, and by cancer site. This study adds to the limited bulk of information about the health consequences, specifically cancer, in local community members affected by the 9/11 terrorist attacks on the WTC. It highlights the WTC EHC Pan-Cancer Database as an important source of information for further health surveillance and research in the population.

Comment: It would be important to expand more the Materials and Methods section by explaining how information on non-WTC-related risk factors were collected (e.g., BMI, smoking status).

Answer: Detailed information about collected data has been added to the Methods section (Page 4, Line 150).

Comment The authors mentioned three cases of mesothelioma occurred in about 3,000 cancer patients with about 18 years of follow-up. Despite the small number it is important and strongly recommended to include description of these cases in the main results rather than in the Supplementary materials.

Answer: Mesothelioma cases information has been added to Table 3 which is the merged form of Table of 3 and 4 in the previous version.      

Comment: The authors mentioned that information on clinically significant biomarkers is collected but they didn’t present any descriptive results in the manuscript, as well as there are no distinct results on cancer patients with multiple primary malignancies.

Answer: This is the first general descriptive study and the data gathering and analysis of biomarkers on the individual cancers are still ongoing and will be the focus of separate analysis and publication. We have modified sentences about biomarkers to prevent misunderstanding.

Comment: The presented results do not provide sufficient ground for the subsequent detailed discussion on possible carcinogenic mechanisms as a result of WTC exposure in this population, this specific paragraph is very speculative and to reviewer’s opinion is not very relevant to the discussion of results. At this stage, the manuscript provides only some basic descriptive characteristics, it would be very interesting and important to see further advanced cancer data statistical analyses in the affected local community members.    

Answer: The sentence about the possible carcinogenic mechanisms have been removed and relevant paragraph has been modified.

Specific comments:

Comment: Results, Table 1: Why the characteristics are given for all 2,840 cancer patients although in the preceding para (p.3 lines 122-124) authors stated that 279 patients with non-melanoma skin cancer were excluded for this analysis?

Answer: This has been corrected throughout the manuscript. Table 1 has been revised.

Comment: Results, Table 1: Please indicate when the BMI measurements were performed, especially in cancer patients as their diseases could directly affect the BMI.

Answer: All demographic information was obtained at the enrollment of the WTC EHC program. Description about this has been added to Page 4, Line 150.

Comment: Results, Table 1: Same question for smoking status, at what moment of time? Cancer diagnosis could probably be the reason of a higher proportion of former smokers compared to cancer-free group.

Answer: Smoking status has been recorded during first enrollment at initial visit. Explanation about this has been added to Page 4, Line 150.

Comment: P.4, line 143: please clarify if all five cancer diagnoses in one patient were primary cancers.

Answer: All of the multiple cancers reported are primary cancers. Explanation about this has been included in the paper, Page 5, Line 203.

Comment: Figure 2. The number of myeloma cases on that figure (n=101) is different from the number of 104 myelomas in the following figures and tables, please check and correct.

Answer: The number should be 104. It has been corrected in Figure 2.

Comment: P.7, line 172, Title of Figure 5: it is suggested to change “blood cancer” for “cancers of lymphoid and haematopoietic tissue” according to ICD terminology and nomenclature. 

Answer: It has been changed as “cancers of lymphoid and hematopoietic tissue” throughout the paper.   

Comment: Tables 3 and 4: please include characteristics for mesothelioma cases. It would be interesting and important to see in the main results sex, age, and latency characteristics of the three cases taking into account asbestos exposure in the study population.

Answer: Table 3 and 4 has been merged and mesothelioma cases have been added.

Round 2

Reviewer 3 Report

Thank you for taking into account the comments; this was appreciated. You will find some comments below.

Notes: the line numbers indicated in the response to authors document have not been updated and were discordant compare to the line numbers of the manuscript, making the checking complicated.

Comments:

L189-191: you have clarified that children could also be enrolled in the WTC EHC; however, I believe that the descriptive analysis that you have performed did not include children under 18 during the study period. Could you clarify this aspect?

L200-202: there is some redundancy with line 228-230; should be kept in the method section.

L203: I would advise to add a section specific to the exposure assessment in the method section, as dust exposure is not a clinical or cancer characteristic.

L225-226: If I understand correctly, dust could exposure information as you’ve defined it comprises acute, chronic, and acute/chronic exposures? Could you clearly mention that you have summarized the information in one exposure variable? Why haven’t you kept the original exposure variables?

L229: typo

L246: the ongoing analysis of multiple primary cancers, cancer risk factors and biomarkers information could be mentioned in the manuscript.

L250-253: you have mentioned in the response to authors document that cancer cases were incident; meaning that the subjects were free of cancer at enrolment in the WTC EHC, and then developed the illness and were considered as cancer patients if they had a certifiable disease. But you also have mentioned that all subjects enrolled in the WTC EHC are patients. Which statement is right? This need to be clarified.

Later in the document, you stated that some patients enrolled because of aerodigestive or mental health disorders have developed cancers. Can you specify that those patients are not accounted twice, as “non-cancer patients” and “cancer patients”, in your analysis?

L264 to 272 should be moved to the analysis section.

L281: I would suggest “demographic characteristics at enrolment”

L281-282: Why haven’t you provided p-value of statistical tests for the cancer/non-cancer comparison?

L325-327: this is redundant with L624-266. Should be kept in the methods (except for numbers).

L335-336: it is stated that 50% of participants reported having being exposed to the WTC dust cloud (acute exposure). If I understand correctly, this variable represents the exposure on 9/11 and not after, is that right? If yes, I suggest to specify it in table 1 and methods.

L361: I suggest to indicate that demographic characteristics described were recorded at enrolment in title of table 1.

Table 1:

- The range of age on 9/11 starts from 0.00 year for both groups, is that correct?

- For Age on 9/11 categorised, I would suggest to add the minimum age of the first category: is it 18-20?

Table 3:

- Number of decimals should be standardized.

L425-426: Interpretation of results should be moved to the discussion section (mesotheliomas and asbestos).

L498-499: I would advise to be careful with this interpretation. You haven’t conducted the right epidemiological study to stipulate that WTC dust cloud exposure may not be a cancer risk factor. Moreover, no statistical tests have been provided, exposure information is not detailed and you compare very different types of cancers as one group. I suggest to remove the sentence.

L512-515: the sentence is unclear.

L591: the reference of the SEER data is missing.

Author Response

Response to reviewer

We would like to give our special appreciation to Reviewer 3 for the careful and detailed review of our manuscript entitled “Characteristics of cancer patients in the World Trade Center Environmental Health Center” by Durmus et al. (ID ijerph-900789). We have now responded to all the comments of Reviewer 3 and believe that this response strengthens the manuscript and makes it appropriate for publication.

Reviewer 3

Thank you for taking into account the comments; this was appreciated. You will find some comments below.

Notes: the line numbers indicated in the response to authors document have not been updated and were discordant compare to the line numbers of the manuscript, making the checking complicated.

Response: Apologies for this issue. Due to track changes, line numbers change automatically in different views. We will be more careful with this and suggest to refer to the line numbers in the “No Markup” mode.

Comments:

Comment:  L189-191: you have clarified that children could also be enrolled in the WTC EHC; however, I believe that the descriptive analysis that you have performed did not include children under 18 during the study

Response: Our descriptive analysis included all patients enrolled in the WTC EHC also children under 18. We apologize if this was not clear and state in the paper.

Any person who has defined WTC exposure and a community member who has the presence of a certifiable WTC-related health condition including students and children under the age of 18 are eligible to be enrolled in the treatment program. There is no minimum age to be enrolled in the program and WTC EHC has a pediatric section and includes children.

The inclusion of children is consistent with previous studies in which we included this age group in a paper entitled “Associations of World Trade Center exposures with pulmonary and cardiometabolic outcomes among children seeking care for health concerns”. To reinforce the presence of children in our population, we have added this reference to the paper (Reference 30). 

We have added the following sentence on Page 3 Line 119:

Thus, enrollees with and without cancers are included, and children under the age of 18 on 9/11/2001 are also eligible [30].

Comment: L200-202: there is some redundancy with line 228-230; should be kept in the method section.

Answer:  Redundancy has been corrected.

Comment: L203: I would advise to add a section specific to the exposure assessment in the method section, as dust exposure is not a clinical or cancer characteristic.

Answer: The reviewer is correct in pointing out that the WTC exposure is not a clinical or cancer characteristic. We have now created a sub-section as ‘2.2. Exposure assessment at WTC EHC’ in the Methods section on Page 3 line 130. 

Comment: L225-226: If I understand correctly, dust cloud exposure information as you’ve defined it comprises acute, chronic, and acute/chronic exposures? Could you clearly mention that you have summarized the information in one exposure variable? Why haven’t you kept the original exposure variables?

Answer: We apologize if we were unclear about our exposure characterizations. Community members had very complex exposures, which have been the source of extensive discussion for many years. Community members may have been in the area on 9/11, and subsequently had massive dust exposure on September 11, 2001 (acute exposure, which we describe as being exposed to the dust clouds on 9/11 (Dust Cloud yes/no)). In addition, community members may have had chronic exposure to resuspended dust as residents – few of whom were evacuated, or as local workers, most of whom returned approximately one week after the disaster to incompletely cleaned areas (defined as chronic exposure). WTC EHC participants also included workers involved in the cleaning of the surrounding area (clean-up workers) and a small number of responders who did not fit the Responder program for a variety of reasons. For our first analysis, we simplified categories of exposure as those with “acute” exposure e.g. they were there on 9/11 and in the dust cloud (Dust Cloud yes/no). We then include the potential for chronic exposure, which depends on the category of activity i.e. local resident, local worker, student, clean-up worker. The acute exposure and the exposure category are not mutually exclusive. As stated, we have used this categorization of exposures in numerous publications. This explanation has been added on Page 3 Line 131.

Comment: L229: typo

Answer: Typo error has been corrected.

 Comment: L246: the ongoing analysis of multiple primary cancers, cancer risk factors and biomarkers information could be mentioned in the manuscript.

Answer: We added a description of these issues in the conclusion section page 13 line 382.

The pathologic and histologic characteristics of cancers with multiple primary cancer analysis for each type of cancer, assessment of cancer risk factors, and cancer-specific biomarker information in our WTC EHC Pan-Cancer database will allow for future analyses of individual and group cancer behavior in this group, as well as long-term follow-up study of this population.

Comment: L250-253: you have mentioned in the response to authors document that cancer cases were incident; meaning that the subjects were free of cancer at enrolment in the WTC EHC, and then developed the illness and were considered as cancer patients if they had a certifiable disease. But you also have mentioned that all subjects enrolled in the WTC EHC are patients. Which statement is right? This need to be clarified.

Later in the document, you stated that some patients enrolled because of aerodigestive or mental health disorders have developed cancers. Can you specify that those patients are not accounted twice, as “non-cancer patients” and “cancer patients”, in your analysis?

Answer: For this analysis, we include patients with a diagnosis of cancer who were enrolled in the WTC EHC between May 2002 and December 31, 2019. For reference to sampling, we also included WTC EHC participants who had signed research consent and had a medical condition that did not include a diagnosis of cancer. These non-cancer participants were enrolled at one clinical site (Bellevue Hospital), during the same period and were used to compare cancer and non-cancer participants to understand if there were basic characteristics or exposures that differed between these groups. Although most patients with cancers were enrolled in the WTC EHC with cancer as their certifying illness, however, some patients developed incident cancers after enrollment for a non-cancer condition.

This information has been added on Page 4 line 183.

If a patient enrolled in the program because of any other health condition other than cancer developed any type of certifiable cancer, this patient has been included in the cancer group and excluded from the non-cancer group for this study. Thus, no patient could be counted twice for this study.

Comment: L264 to 272 should be moved to the analysis section.

Answer: We appreciate the reviewer’s comment. Our intent was to include only the analytic methods that we have used in the methods section and thus we would like to keep this section in the results section to allow for easy understanding of Figure 1. We have, however, also revised the analysis section accordingly.

 Comment: L281: I would suggest “demographic characteristics at enrolment”

Answer: The legend for Table 1 has been changed as “Descriptive characteristics of patients with/without cancers at enrollment to the WTC EHC (non-melanoma skin cancers excluded).”

Comment: L281-282: Why haven’t you provided p-value of statistical tests for the cancer/non-cancer comparison?

Answer: This question raises a very important and complex point about the use of p values. As is now well known, high impact medical and public health journals mostly require papers to report how the study subjects are selected into the study. This paper is to provide descriptive characteristics of cancers at WTC EHC, and we include the non-cancer subjects to make it clear how the cancer patients from WTC EHC were selected into the study. Univariate comparison between the heterogeneous cancer patients with highly heterogeneous non-cancer subjects would not be meaningful and could be misleading without suitable adjustment for various confounding factors (e.g. smoking for lung cancers). Formal multivariable evaluation and comparison of cancer and non-cancer subjects are important future research topics that are beyond the scope of the current manuscript which focuses on descriptive characteristics of cancers at WTC EHC.  

 Comment: L325-327: this is redundant with L624-266. Should be kept in the methods (except for numbers).

Answer: Both information has been moved to the Analysis subsection of the Methods section.

Comment:  L335-336: it is stated that 50% of participants reported having being exposed to the WTC dust cloud (acute exposure). If I understand correctly, this variable represents the exposure on 9/11 and not after, is that right? If yes, I suggest to specify it in table 1 and methods.

Answer: Yes, that is correct. As defined in methods, we defined “dust cloud” exposure as exposure to the heavy dust from the dust cloud on the day of 9/11/2001. The sentences about dust cloud have been revised as following:

Community members may have been in the area on 9/11, and subsequently had massive dust exposure on September 11, 2001 (acute exposure, which we describe as being exposed to the dust clouds on 9/11 (Dust Cloud yes/no)). In addition, community members may have had chronic exposure to resuspended dust as residents – few of whom were evacuated, or as local workers, most of whom returned approximately one week after the disaster to incompletely cleaned areas (defined as chronic exposure).

Comment: L361: I suggest to indicate that demographic characteristics described were recorded at enrolment in title of table 1.

Answer: The title of Table 1 has been revised as “Descriptive characteristics of patients with/without cancers at enrollment to the WTC EHC (non-melanoma skin cancers excluded).

 Table 1:

Comment: The range of age on 9/11 starts from 0.00 year for both groups, is that correct?

Answer: Yes, that is correct as some women were pregnant on the day of 9/11 disaster. Their babies were eligible to be enrolled in the pediatric section of the WTC EHC [29]. Thus, their age has been accepted as 0 on the day of the disaster.

Comment:  For Age on 9/11 categorised, I would suggest to add the minimum age of the first category: is it 18-20?

Answer: There is no minimum age to be enrolled in the program. We have revised the following sentences on Page 3 line 119 as following:

In addition, the enrollee must have a certifiable WTC-related health condition that includes an aerodigestive disorder, mental health symptoms, consistent with PTSD, depression or anxiety, or cancer [29]. Thus, enrollees with and without cancers are included, and children under the age of 18 on 9/11/2001 are also eligible [30].

We have also cited a pediatric publication from our clinic (Reference 30).

 Table 3:

Comment: Number of decimals should be standardized.

Answer: The number of decimals has been standardized.

Comment: L425-426: Interpretation of results should be moved to the discussion section (mesotheliomas and asbestos).

Answer: The discussion part has been deleted from the Results section as suggested.

Comment: L498-499: I would advise to be careful with this interpretation. You haven’t conducted the right epidemiological study to stipulate that WTC dust cloud exposure may not be a cancer risk factor. Moreover, no statistical tests have been provided, exposure information is not detailed and you compare very different types of cancers as one group. I suggest to remove the sentence.

Answer: Thanks to the Reviewer for this suggestion. We agree, the sentence has been removed.

Comment: L512-515: the sentence is unclear.

Answer: The unclear sentence has been revised as following:

Multiple primary cancers were reported for thyroid cancer in responders [26] which is consistent with our data.

Comment: L591: the reference of the SEER data is missing.

Answer: Reference for SEER data has been added.